# The stigma associated with cutaneous leishmaniasis (CL) and mucocutaneous leishmaniasis (MCL): A protocol for a systematic review

Hasara Nuwangi[1], Thilini Chanchala Agampodi[1], Helen Philippa Price[2], Thomas Shepherd[3], Kosala Gayan Weerakoon[4], Suneth Buddhika Agampodi[5,6] *

1 Department of Community Medicine, Faculty of Medicine and Allied Sciences, Rajarata University of Sri Lanka, Anuradhapura, Sri Lanka, 2 School of Life Sciences, Keele University, Newcastle-under-Lyme, Staffordshire, United Kingdom, 3 School of Medicine, Keele University, Newcastle-under-Lyme, Staffordshire, United Kingdom, 4 Department of Parasitology, Faculty of Medicine and Allied Sciences, Rajarata University of Sri Lanka, Anuradhapura, Sri Lanka, 5 Department of Internal Medicine, Section of Infectious Diseases, Yale University School of Medicine, New Haven, Connecticut, United States of America, 6 International Vaccine Institute, Seoul, Republic of Korea

☯ These authors contributed equally to this work.
* sunethagampodi@yahoo.com

**Data Availability Statement:** No datasets were generated or analysed during the current study. All

## Abstract

Leishmaniasis is a neglected tropical disease with three main clinical types; cutaneous leishmaniasis (CL), mucocutaneous leishmaniasis (MCL), and visceral leishmaniasis (VL). CL and MCL are considered to be highly stigmatizing due to potentially disfiguring skin pathology. CL and MCL-associated stigma are reported across the world in different contexts assimilating different definitions and interpretations. Stigma affects people with CL, particularly in terms of quality of life, accessibility to treatment, and psycho-social well-being. However, evidence on CL- and MCL-associated stigma is dispersed and yet to be synthesized. This systematic review describes the types, measurements, and implications of the stigma associated with CL and MCL and identifies any preventive strategies/interventions adopted to address the condition. This study was developed according to the Preferred Reporting Items for Systematic Reviews and Meta-Analysis Protocols (PRISMA-P) statement which is registered in the International Platform of Registered Systematic Review and Meta-analysis Protocols PROSPERO (ID- CRD42021274925). We will perform an electronic search in MEDLINE, Embase, Scopus, PubMed, EBSCO, Web of Science, Global Index Medicus, Trip, and Cochrane Library databases, and in Google Scholar, using a customized search string. Any article that discusses any type of CL- and/or MCL-associated stigma in English, Spanish and Portuguese will be included. Articles targeting veterinary studies, sandfly vector studies, laboratory-based research and trials, articles focusing only on visceral leishmaniasis, and articles on diagnostic or treatment methods for CL and MCL will be excluded. Screening for titles and abstracts and full articles and data extraction will be conducted by two investigators. The risk of bias will be assessed through specific tools for different study types. A narrative synthesis of evidence

relevant data from this study will be made available upon study completion.

**Funding:** The ECLIPSE program is funded by the National Institute for Health and care Research (NIHR) (NIHR200135) using UK aid from the UK Government to support global health research. The views expressed in this article are those of the authors and not necessarily those of the NIHR or the UK Department of Health and Social Care.

**Competing interests:** The authors have declared that no competing interests exist.

will then follow. This review will identify the knowledge gap in CL-associated stigma and will help plan future interventions.

## Introduction

Leishmaniasis is a tropical parasitic disease endemic in 98 countries across the globe including parts of Southern Europe, the Middle East, Africa, Asia, and South America [1]. The disease affects around 700,000–1,000,000 people every year [2]. Leishmaniasis is caused by a protozoan parasite, *Leishmania spp*. It is transmitted to humans by the bite of a female sandfly (genera *Phlebotomus* and *Lutzomyia*). Cutaneous leishmaniasis (CL), mucocutaneous leishmaniasis (MCL), and visceral leishmaniasis (VL) are the three main clinical types of this disease. CL is characterized by ulcerated skin lesions or non-ulcerative nodules. The size and the severity of the lesions vary [1]. MCL can be present as various manifestations such as lesions in the nasal and oral cavity, lesions in the subglottic region, trachea, and vocal cords. Some might even lead to the destruction of face structure, and in extreme cases even death [3, 4]. The prevalence of CL and MCL is increasing in many regions of the world, including Europe, the Middle East, Northern Africa, Asia, and parts of South America, and is recognized as a disease of the poor [5–8]. As these two manifestations affect bodily appearance, the stigma related to these conditions is likely to differ from that for visceral leishmaniasis.

### Definitions and dimensions and mechanisms of stigma

Ervin Goffman pioneered research on stigma. In his book Stigma: Notes on the management of spoiled identity, 1963 he describes stigma as "the situation of the individual who is disqualified from full social acceptance" [9]. Stigma is defined in various ways, across disciplines. In their book: The Dilemma of Difference, researchers Ainlay, Becker, and Coleman propose a new definition of stigma as "a characteristic of persons that is contrary to a norm of a social unit. The characteristic may involve what people do (or have done), what they believe or whom they are (owing to physical or social characteristics)" [10]. Katz, in 1982, explains that "Stigmatized individuals have attributes that do not accord with the prevailing standards of the normal and good. They are often denigrated and avoided openly in the case of known criminals and other transgressors, or covertly and even unconsciously" [11]. In 2001, Link and Phelan discuss reasons for the variations in the definition of stigma; including the concept being applied to a multitude of unique circumstances, research on stigma being multidisciplinary, and researchers approaching the concept of stigma from different theoretical angles leading to different conceptualizations [12].

Not only the definition but also the classification and types of stigma vary according to the discipline of study and the disease under investigation [12]. Weiss describes three major types of stigma as enacted, anticipated, and internalized related to neglected tropical diseases (NTDs) [13]. Scambler and Hopkins define felt stigma as refers principally to the fear of enacted stigma. "a feeling of shame associated with a particular disease". According to them enacted stigma is "instances of discrimination against people on the grounds of their perceived unacceptability or inferiority" [14]. Weiss defines Internalized stigma as "when a person with a stigmatized condition accepts perceived exclusionary views of society and self-stigmatizes himself or herself" [13]. With regard to tuberculosis, six types of stigma have been put forward: public stigma, anticipated or perceived stigma, secondary stigma, internalized or self-stigma, enacted or experienced stigma, and structural stigma [15].

## Health-related stigma

Health-related stigma is described as the "social disqualification of individuals and populations who are identified with particular health problems" [16]. Health-related stigma leads to poor quality of life and mental health issues are a challenge to public health and a barrier to health interventions. Stigma can lead to poor prognosis, increased risk of disability, delay in diagnosis and treatment, and continuing risk of disease transmission, resulting in increased morbidity as well as psychological problems such as depression, stress, and fear, [16–18]. Link, Phelan, and Hatzenbuehler argue that stigma should be considered a fundamental cause of health inequalities [19]. Structural social determinants such as racism, gender, and poverty can compound and aggravate stigma and restrict access to healthcare leading to inequities in health [20]. When socio-economics, politics, and inequalities that emerge from those structures are ignored, physicians can potentially misdiagnose patients and do real measurable harm [21]. Many studies identify social factors as the root of health inequalities. This highlights the importance of understanding and measuring stigma to assess the burden and implications of a particular disease and inform health system policy recommendations [6]. Furthermore, Hatzenbuehler emphasizes the need for more research to link measures of structural stigma to individual levels. As structural stigma is complex and needs an interdisciplinary approach to address, he emphasizes the need for future research that incorporates different fields such as sociology, anthropology, and epidemiology, with diverse methods and approaches [19, 22].

## CL-associated stigma

Several studies have investigated the stigma associated with CL and how it disproportionately affects socio-economically deprived populations including women and children [23, 24]. Diverse overlapping classifications are available for CL-associated stigma. According to Al-Kamel, three types of CL-associated stigma are prevalent in society; CL-related social stigma, CL aesthetic stigma, and CL psychological stigma [25]. CL-related social stigma leads to the negative beliefs and perceptions that society has about people with the condition, whereas CL aesthetic stigma is explained as the dissatisfaction occurring from changes to the bodily image (scars, lesions, deformities) resulting from the condition. CL psychological stigma is the deep psychological stress that arises from experiencing the CL-related social stigma, CL aesthetic stigma [25]. Other potential types of stigma commonly associated with skin-related NTDs are likely to exist among CL patients and need to be further explored, including felt stigma, the expectation of discrimination because of CL; enacted stigma, and the unfair treatment from others towards CL patients [26–28]. Stigma does not affect everyone equally and depends on a range of factors. There is a gap in knowledge around stigma associated with CL and MCL in particular when compared to other diseases such as leprosy and tuberculosis [15, 29, 30].

## Measuring stigma

Although many interventions have been conducted to reduce health-related stigma, the effectiveness of these remains uncertain [16]. One explanation is the lack of validated tools to evaluate the impact of such interventions and accurately measure changes in the levels of stigma before and after the interventions. Link et al. emphasized the need for well-developed instruments to carry out stigma-related research. The observation and measurement of stigma are essential for the scientific understanding of the concept [31] and the need for in-depth studies on the stigma associated with CL is highlighted in the literature [32]. While mental health effects and the psychosocial burden of CL have been reviewed [32–34], evidence synthesis focused on the stigma associated with CL and MCL and its implications as a whole has not

been generated to date. This study aims to systematically review the dimensions, methods of measurement, implications, and preventive strategies of the stigma associated with CL.

## Methods

This study has been registered on the International Platform of Registered Systematic Review and Meta-analysis Protocols (PROSPERO) under the registration number CRD42021274925. The systematic review protocol is prepared according to the guidelines given by the Preferred Reporting Items for Systematic Review and Meta-Analysis Protocols (PRISMA-P) statement [35, 36] and the checklist is presented (S1 File).

### Objectives

This review aims to systematically explore the stigma associated with CL and MCL and to identify related knowledge gaps.

This review seeks to address the following questions:

1. What are the types of stigma associated with CL and MCL?

2. What are the means of measuring stigma associated with CL and MCL and the different tools used?

3. What are the implications of the stigma associated with CL and MCL, on patients and their families?

4. What are the preventive strategies/interventions developed to address the stigma associated with CL and MCL?

### Search strategy

A systematic search of published literature will be conducted using electronic databases, namely; MEDLINE, Embase, Scopus, PubMed, EBSCO, Web of Science, Global Index Medicus, Cochrane Library databases, and Google Scholar. Articles up to January 2023 will be considered in the search. The reference lists of included studies will be hand-searched to identify any additional evidence and potential sources of information. The articles will be searched using titles, abstracts, and keywords. The search strategy will be built under two main search term categories; terms describing stigma and terms describing the disease. The keywords were selected based on literature and after consulting the experts in leishmaniasis both locally and internationally.

Stigma*, discriminat*, stereotyp*, negative attitude, psychosocial impact, psychosocial burden, social consequences, stigmatizing effect, scar*, disfigure*, self-stigma, label avoidance, disgrace, shame, perception, and rejection

The keywords to select CL-related articles will be:

Dermal leishman*, cutaneous leishman*, oriental sore, Uta, Chiclero ulcer, tropical sore, Bagdad boil, Baghdad boil, Bauer ulcer, Delhi boil, Aleppo boil, Aleppo button, Jericho boil, one year sore, one year ulcer, tegumentary leishmaniasis, Biskra button, Biskra nodule, Calcutta ulcer, Jericho button, Kandahar sore, Lahore sore, Oriental button, Pian bois and Old World leishmaniasis.

The keywords to select MCL-related articles will be: Mucosal Leishman*, Mucocutaneous Leishman*, muco cutaneous Leishman*, espundia, nasopharyngeal Leishman*, New World leishmaniasis, American leishmaniasis, leishmaniasis americana

The final search string will be created as a combination of these key term categories. PubMed search results are provided as supporting documents (S2 File).

Two independent reviewers will carry out the whole process of systematic searches. Any disagreements will be solved by a discussion with the help of a third reviewer. A consistent decision will be reached after a discussion.

**Inclusion criteria.**

- All articles which discuss any type of CL- and MCL-associated stigma entirely or partially

- Articles in English, Spanish and Portuguese only

- Qualitative studies, quantitative studies, cross-sectional studies, mixed-method studies, cross-sectional, grounded theory, ethnographic/anthropological, survey, case studies, and multidisciplinary studies will be included

- Original articles, literature reviews, workshop abstracts, conference abstracts, conference presentations

**Exclusion criteria.**

- Articles targeting veterinary studies, vector studies, laboratory-based research, and trials

- Articles that explore stigma only in visceral leishmaniasis

- Articles focused on diagnostic or treatment methods for CL and MCL

- Articles which are focused on skin lesions caused by PKDL

## Study flow and data management

First, duplications will be removed from the search results and the studies to be included will be first screened by title, abstract, and keywords. The investigators will review titles and abstracts to remove the obviously irrelevant studies and later in the second screening stage, full texts will be screened. The reason for the exclusion of articles retrieved will be documented. Then the full text will be screened and articles will be selected. Two reviewers will conduct the whole process independently and any disagreements will be resolved through discussion with a third reviewer.

## Data extraction

A pre-piloted data extraction form will be used to extract data from the full text of all screened eligible articles. The data extraction form will include:

1. Title

2. Authors

3. Year of publication

4. Country/region

5. Study setting and study design

6. Reported demographics of participants

7. Time period

8. Method of data collection

9. Stigma type

10. Stigma scales used

11. Characteristics of scales used

12. Validity and reliability of scales

13. Accuracy parameters of the scales used

14. Any other conceptualization of stigma used

15. Outcomes and impacts of the study

16. Limitations of the study

17. The conclusion of the study

18. Recommendations/Future directions

Two individual investigators will collect data independently. If there is missing information, the investigators will contact the corresponding authors of those particular studies through the correspondents' email provided within the article. Any contradictions in the data extraction will be resolved by discussion with the help of another author, and a final consistent decision will be made. The data extraction form is displayed as supporting documents (S3 File).

## Data synthesis

Since the review uses both quantitative and qualitative studies, the analysis will accommodate both quantitative and descriptive approaches. Summary tables of all studies that are included will be produced including information such as title, publication year, the country in which the study was conducted, and a summary of findings. A narrative synthesis will be completed using the integrative review method. This method is used to summarize theoretical and empirical literature to convey a thorough understanding of a studied problem [37]. Due to the nature of the review questions, heterogeneity of included articles is expected. However, since a meta-analysis would not be conducted and the findings will be described for each review question separately as mentioned above this heterogeneity will not account for an error. Investigators will pay attention to heterogeneity in interpreting the findings in the narrative synthesis of evidence and discuss the evidence acknowledging the heterogeneity expected.

## Assessment of risk of bias

Study quality of observational cohorts, cross-sectional studies, case-control studies, case series studies, and controlled intervention studies will be assessed by the study quality assessment toolset of the National Heart, Lung and Blood Institute (Study Quality Assessment Tools | NHLBI, NIH, 2021). The risk of bias in randomized control trials will be assessed by the revised Cochrane tool for assessing the risk of bias in randomized control trials (RoB tool) [38, 39]. All the other types of studies will be assessed using the Mixed Methods Appraisal Tool (MMAT). The MMAT was developed to appraise the quality of empirical studies and allows assessment of the methodological quality of quantitative, qualitative, and mixed methods studies [39]. We will assess qualitative studies for credibility, transferability, dependability and confirmability [40]. The study quality of each eligible study will be examined by two independent authors. The authors will indicate the appropriateness of the study for particular criteria, required by the tools. If there's a major flow in any of the criteria that will be noted and later discussed. After individual assessments, each author will give a score to each article and then a final score will be decided after discussion. Any queried articles will be reviewed by a third author and a final consistent decision will be made. We will present the identified strengths

and weaknesses with regard to the quality of the study for the selected studies for synthesis and will discuss the implications of the quality of studies in the generation of evidence with regard to stigma in CL and MCL [41].

## Discussion

The topic of stigma is increasingly being considered across disciplines such as public health, mental health, medical anthropology, and sociology. However, stigma related to NTDs is relatively under-researched, especially for emerging infectious diseases such as CL and MCL in countries like Sri Lanka. CL-associated stigma is poorly understood, which hinders the ability to produce interventions to reduce stigma or address stigma-related issues. It is important to understand the existing knowledge on the stigma associated with CL as well as the implications of stigma on CL patients' and their family members' lives. Identification of the impacts of CL-associated stigma will be critical to informing the development of future preventive strategies and interventions. A review of the tools available to measure stigma will facilitate the development process of a gold standard tool to assess stigma in CL and similar NTDs. A narrative synthesis of different aspects of stigma related to the disease will be helpful in conceptualizing the concept of stigma itself and also identifying the drivers of stigma in CL and MCL informing public health experts and policymakers to plan appropriate interventions. This review will also highlight the geographical variations and commonalities of stigma in CL which will be helpful in planning global health interventions to reduce stigma and the psychosocial burden related to CL and MCL. In this study, we are not conducting a meta-analysis. Given the anticipated nature of the heterogeneity of the articles, tools, cultural influences and subjective nature of the concept we believe it will be unfeasible to conduct a meta-analysis. Another limitation of the study is that we will be not including publications in conference proceedings and other grey literature due to time constraints and lack of practicality. There could be other articles from other languages beyond Spanish, Portuguese and English which will not be used in this study which is another limitation.

This review will collate the knowledge on the stigma associated with CL from across the world and will be pivotal to understanding the existing evidence gaps for further exploration, and for the development and implementation of interventions to reduce stigma and related implications.

## Supporting information

**S1 File. PRISMA-P checklist.**
(PDF)

**S2 File. Search conducted in PubMed on 17th January 2023.**
(PDF)

**S3 File. Form to extract data from included studies.**
(PDF)

## Author Contributions

**Conceptualization:** Hasara Nuwangi, Thilini Chanchala Agampodi, Helen Philippa Price, Thomas Shepherd, Kosala Gayan Weerakoon, Suneth Buddhika Agampodi.

**Funding acquisition:** Helen Philippa Price, Suneth Buddhika Agampodi.

**Methodology:** Hasara Nuwangi, Thilini Chanchala Agampodi, Helen Philippa Price, Thomas Shepherd, Kosala Gayan Weerakoon, Suneth Buddhika Agampodi.

**Project administration:** Suneth Buddhika Agampodi.

**Supervision:** Thilini Chanchala Agampodi, Helen Philippa Price, Thomas Shepherd, Kosala Gayan Weerakoon, Suneth Buddhika Agampodi.

**Writing – original draft:** Hasara Nuwangi.

**Writing – review & editing:** Hasara Nuwangi, Thilini Chanchala Agampodi, Helen Philippa Price, Kosala Gayan Weerakoon, Suneth Buddhika Agampodi.

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
