## [Decision Letter · Decision Letter 0]

6 Jan 2023

PONE-D-22-13301The stigma associated with cutaneous (CL) and mucocutaneous leishmaniasis (MCL): a protocol for a systematic reviewPLOS ONE

Dear Dr. Suneth B Agampodi,

Thank you for submitting your manuscript to PLOS ONE. After careful consideration, we feel that it has merit but does not fully meet PLOS ONE’s publication criteria as it currently stands. Therefore, we invite you to submit a revised version of the manuscript that addresses the points raised during the review process.

We look forward to receiving your revised manuscript.

Kind regards,

Alireza Badirzadeh

Academic Editor

PLOS ONE

Journal Requirements:

Reviewers' comments:

Reviewer's Responses to Questions

**Comments to the Author**

1. Does the manuscript provide a valid rationale for the proposed study, with clearly identified and justified research questions?

Reviewer #1: Yes

Reviewer #2: Yes

2. Is the protocol technically sound and planned in a manner that will lead to a meaningful outcome and allow testing the stated hypotheses?

Reviewer #1: Yes

Reviewer #2: Partly

3. Is the methodology feasible and described in sufficient detail to allow the work to be replicable?

Reviewer #1: Yes

Reviewer #2: Yes

4. Have the authors described where all data underlying the findings will be made available when the study is complete?

Reviewer #1: Yes

Reviewer #2: Yes

5. Is the manuscript presented in an intelligible fashion and written in standard English?

Reviewer #1: Yes

Reviewer #2: Yes

6. Review Comments to the Author

You may also provide optional suggestions and comments to authors that they might find helpful in planning their study.

Reviewer #1: The prevalence of MCL and CL leishmaniasis is high in South America and a lot of articles from that area are in Portuguese and Spanish. Similarly a lot of articles of middle east (where CL is common) are not in English. I recommend to include articles of Spanish and Portuguese at least and use google translator to retrieve data from those articles.

Reviewer #2: This article is an interesting topic, but there are several comments that must be addressed:

-Since cutaneous leishmaniasis has local and regional names, it is necessary to consider these names in the search strategy, including: Biskra button and Biskra nodule; Calcutta ulcer; Jericho button; Kandahar sore; Lahore sore; Oriental button; Pian bois and old World leishmaniasis.

- If the articles which skin lesions caused by the PKDL and para KDL are not included, please mention it in the exclusion criteria section.

- If the authors will be used MeSH (Medical Subject Headings) database, please declare it in the text of the protocol.

- The year and month of the start and end of the search are not clear in the article.

- There is no PRISMA flowchart in the article.

- The Discussion section is very small and the limitations of this protocol are not known.

-Please state that have consulted with experts in this field to select keywords and syntax?

-Please clarify the contact way with the corresponding authors, if there is missing information.

-How do you justify the heterogeneity of included studies?

-In the section on Risk of bias, please mention the quality appraisal tool in detail (scoring,…).

7. PLOS authors have the option to publish the peer review history of their article (what does this mean?). If published, this will include your full peer review and any attached files.

Reviewer #1: **Yes: **Shahnewaj Bin Mannan

Reviewer #2: No

---

## [Author Response · Author response to Decision Letter 0]

2 Feb 2023

Reviewer #1 comments

Comment - The prevalence of MCL and CL leishmaniasis is high in South America and a lot of articles from that area are in Portuguese and Spanish. Similarly a lot of articles of middle east (where CL is common) are not in English. I recommend to include articles of Spanish and Portuguese at least and use google translator to retrieve data from those articles.

Response - We thank the reviewer for this valid comment. We accept this comment and the manuscript was changed to reflect this (line 212)

Reviewer #2 comments

Comment – This article is an interesting topic, but there are several comments that must be addressed:-Since cutaneous leishmaniasis has local and regional names, it is necessary to consider these names in the search strategy, including: Biskra button and Biskra nodule; Calcutta ulcer; Jericho button; Kandahar sore; Lahore sore; Oriental button; Pian bois and old World leishmaniasis.

Response - Thank you, reviewer, for this comment. This comment is very well received. We changed the search string by adding these suggested terms (line 198-200)

Comment – If the articles which skin lesions caused by the PKDL and para KDL are not included, please mention it in the exclusion criteria section.

Response- This was corrected (line 222)

Comment – If the authors will be used MeSH (Medical Subject Headings) database, please declare it in the text of the protocol.

Response - We are not using MeSH databases in this review hence we did not mention about the MeSH database

Comment – The year and month of the start and end of the search are not clear in the article.

Response- This was rewritten to convey the idea more comprehensively (line 185-186)

Comment – There is no PRISMA flowchart in the article.

Response - PRISMA-P Checklist was submitted as a supporting document (S1 Checklist) as per journal requirements

Comment – The Discussion section is very small and the limitations of this protocol are not known.

Response - The discussion section was improved and limitations were discussed as the reviewer proposed. Line- 302-313

Comment – Please state that have consulted with experts in this field to select keywords and syntax?

Response - Thank you for this comment. This was included in the manuscript (line 189-190)

Comment – Please clarify the contact way with the corresponding authors, if there is missing information

Response - This was modified in the text 219-221

Comment – How do you justify the heterogeneity of included studies?

Response - This was included line 262-268

Comment – In the section on Risk of bias, please mention the quality appraisal tool in detail (scoring,…)

Response - This was corrected (Line 277-286)

---

## [Decision Letter · Decision Letter 1]

28 Apr 2023

The stigma associated with cutaneous leishmaniasis (CL) and mucocutaneous leishmaniasis (MCL): a protocol for a systematic review

PONE-D-22-13301R1

Dear Dr. Suneth B Agampodi,

We’re pleased to inform you that your manuscript has been judged scientifically suitable for publication and will be formally accepted for publication once it meets all outstanding technical requirements.

Kind regards,

Alireza Badirzadeh

Academic Editor

PLOS ONE

Additional Editor Comments (optional):

Reviewers' comments:

Reviewer's Responses to Questions

**Comments to the Author**

1. Does the manuscript provide a valid rationale for the proposed study, with clearly identified and justified research questions?

Reviewer #2: Yes

2. Is the protocol technically sound and planned in a manner that will lead to a meaningful outcome and allow testing the stated hypotheses?

Reviewer #2: Yes

3. Is the methodology feasible and described in sufficient detail to allow the work to be replicable?

Reviewer #2: Yes

4. Have the authors described where all data underlying the findings will be made available when the study is complete?

Reviewer #2: Yes

5. Is the manuscript presented in an intelligible fashion and written in standard English?

Reviewer #2: Yes

6. Review Comments to the Author

You may also provide optional suggestions and comments to authors that they might find helpful in planning their study.

Reviewer #2: I have already send my comments and all of the comments were edited and corrected by the authors.

I have already send my comments and all of the comments were edited and corrected by the authors.

7. PLOS authors have the option to publish the peer review history of their article (what does this mean?). If published, this will include your full peer review and any attached files.

Reviewer #2: No

---

## [Editor Report · Acceptance letter]

4 May 2023

PONE-D-22-13301R1 

The stigma associated with cutaneous leishmaniasis (CL) and mucocutaneous leishmaniasis (MCL): a protocol for a systematic review 

Dear Dr. Agampodi:

I'm pleased to inform you that your manuscript has been deemed suitable for publication in PLOS ONE. Congratulations! Your manuscript is now with our production department. 

Kind regards, 

on behalf of

Dr. Alireza Badirzadeh 

Academic Editor

PLOS ONE